# Paninvasion severity assessment of a U.S. grape pest to disrupt the global wine market

Nicholas A. Huron [1]✉, Jocelyn E. Behm[1] & Matthew R. Helmus [1]

Economic impacts from plant pests are often felt at the regional scale, yet some impacts expand to the global scale through the alignment of a pest's invasion potentials. Such globally invasive species (i.e., paninvasives) are like the human pathogens that cause pandemics. Like pandemics, assessing paninvasion risk for an emerging regional pest is key for stakeholders to take early actions that avoid market disruption. Here, we develop the paninvasion severity assessment framework and use it to assess a rapidly spreading regional U.S. grape pest, the spotted lanternfly planthopper (*Lycorma delicatula;* SLF), to spread and disrupt the global wine market. We found that SLF invasion potentials are aligned globally because important viti-cultural regions with suitable environments for SLF establishment also heavily trade with invaded U.S. states. If the U.S. acts as an invasive bridgehead, Italy, France, Spain, and other important wine exporters are likely to experience the next SLF introductions. Risk to the global wine market is high unless stakeholders work to reduce SLF invasion potentials in the U.S. and globally.

[1] Integrative Ecology Lab, Department of Biology, Temple University, Philadelphia, PA 19122, USA. ✉email: nahuron@gmail.com

Invasive plant pests cause substantial economic impacts[1], but most pests and their impacts are confined to specific regions. For a regional pest to become a globally invasive species that disrupt global markets (i.e., paninvasive), ecological and economic factors that determine the pest's transport, establishment, and impact potentials must be aligned at the global scale (Fig. 1, Supplementary methods)[2]. First, paninvasive pests have high transport potential because they can be easily transported among regions, often through global trade[3]. Second, paninvasive pests have high establishment potential, because their environmental needs for population growth are met in many regions[4]. Third, paninvasive pests have high impact potential, because invaded regions have sizeable agricultural production and industries vulnerable to the pest[5]. If these invasion potentials are correlated across multiple regions globally for an emerging regional pest, there is a high risk of the pest spreading to cause supply crashes in regional markets that cascade to disrupt global markets[6].

Despite the importance of identifying emerging paninvasives, existing approaches lack a cohesive and universal framework for rapidly assessing and effectively communicating such risk to stakeholders[7]. To address this gap, we developed the paninvasion severity assessment framework by adapting the U.S. CDC pandemic severity assessment framework to invasion process theory, which describes translocations of species in terms of transport, establishment, and impact potentials (Figs. 1 and 2)[2,8–10]. Although invasive species frameworks are increasingly adapted to understand infectious diseases like COVID-19[11–16], adapting public-health frameworks to invasion science is novel and leverages an increasingly universal risk vocabulary (Fig. 2)[17]. Under this framework, we assessed the paninvasion risk of the spotted lanternfly planthopper (Hemiptera: *Lycorma delicatula*; SLF, Fig. 1). SLF was introduced to South Korea and Japan in the early 2000s and then to the U.S. (ca. 2014) on goods imported from its native China[18]. SLF has rapidly spread from Pennsylvania to several other states, presenting increased opportunities for stepping-stone, bridgehead invasions to additional regions[19–21]. SLF greatly impacts grape production[22–26] and has been presented to the public as one of the worst invasive species to establish in the U.S. in a century[27–29], but its paninvasion risk has not been assessed[21].

SLF likely has high global transport potential because it lays inconspicuous egg masses on plants, stone, and trade infrastructure (e.g., containers, railcars, pallets), which facilitates long-distance transport when eggs are laid on exported items (Fig. 1a–c). Landscaping stone imported from China was the likely vector of the U.S. invasion[18]. Following successful transport, SLF global establishment potential is likely enhanced by the cosmopolitan distribution of its preferred host plant, the tree of heaven (*Ailanthus altissima*, TOH, Fig. 1d–f)[21]. The native ranges of SLF and TOH overlap in China, but for >250 years TOH has escaped cultivation into disturbed habitats and agricultural margins in temperate, subtropical, and Mediterranean regions globally (Fig. 1, map). Once established, SLF likely has a high global impact potential on wine markets because grape is an equally suitable host. SLF develops at similar rates when fed grape or TOH, and fecundity increases when fed a mixed diet of these two preferred hosts[25,30–33]. Asian vineyard production is impacted by SLF infestations[34,35]; and SLF-invaded U.S. vineyards have reported vine deaths, >90% yield losses, and closure (Fig. 1g–i)[21,23,36].

To assess paninvasion risk, we calculated invasion potentials from estimates of SLF transport, establishment, and impact potentials from the U.S. invaded region to uninvaded U.S. states and countries using trade, species distribution models (SDM), and grape and wine production data. We then mapped invasion potentials, calculated alignment correlations, and estimated risk to the $300B global wine market[37].

# Emerging Paninvasive Species
## Spotted Lanternfly
### *Lycorma delicatula*

**Fig. 1 The concept of paninvasive species is based on invasion process theory.** Any species has potential to complete three sequential stages (depicted by the arrow) to become invasive in a region (i.e., transport to the region, establishment in the region, impact on the region's economy)[2]. For a pest paninvasion to occur, a pest must be transported and established in suitable regions globally where it impacts susceptible agriculture, disrupting markets at a global scale (depicted by the bracket). Here, we focus on estimating invasion stage potentials for the spotted lanternfly planthopper (*Lycorma delicatula*, SLF), an emerging U.S. pest at risk of disrupting the global wine market. Photos introduce SLF biology. Gravid females (**a**) lay eggs on many surfaces like stone (**b**) and transport infrastructure (**c**). Once eggs hatch, SLF develop by molting through three black and one red nymphal instars (**d**) before molting into winged adults (top, **a**). SLF feed on phloem of many plants but develop quickly on tree of heaven (*Ailanthus altissima*, TOH, **e**, **f**), a globally invasive tree commonly found in habitat fragments around railroads (**f**) and warehouses (**c**). SLF also heavily feed on grape (**g**), reducing yield (**h**) and contributing to vine death (**i**). Photo credits: S. Cannon (**e**), M. Helmus (top, **a**, **d**), H. Leach (**b**, **c**, **g–i**), G. Parra (**f**). The map is of SLF (orange) and TOH (blue) presences (ca. 2020) we used to estimate paninvasion risk (see Methods).

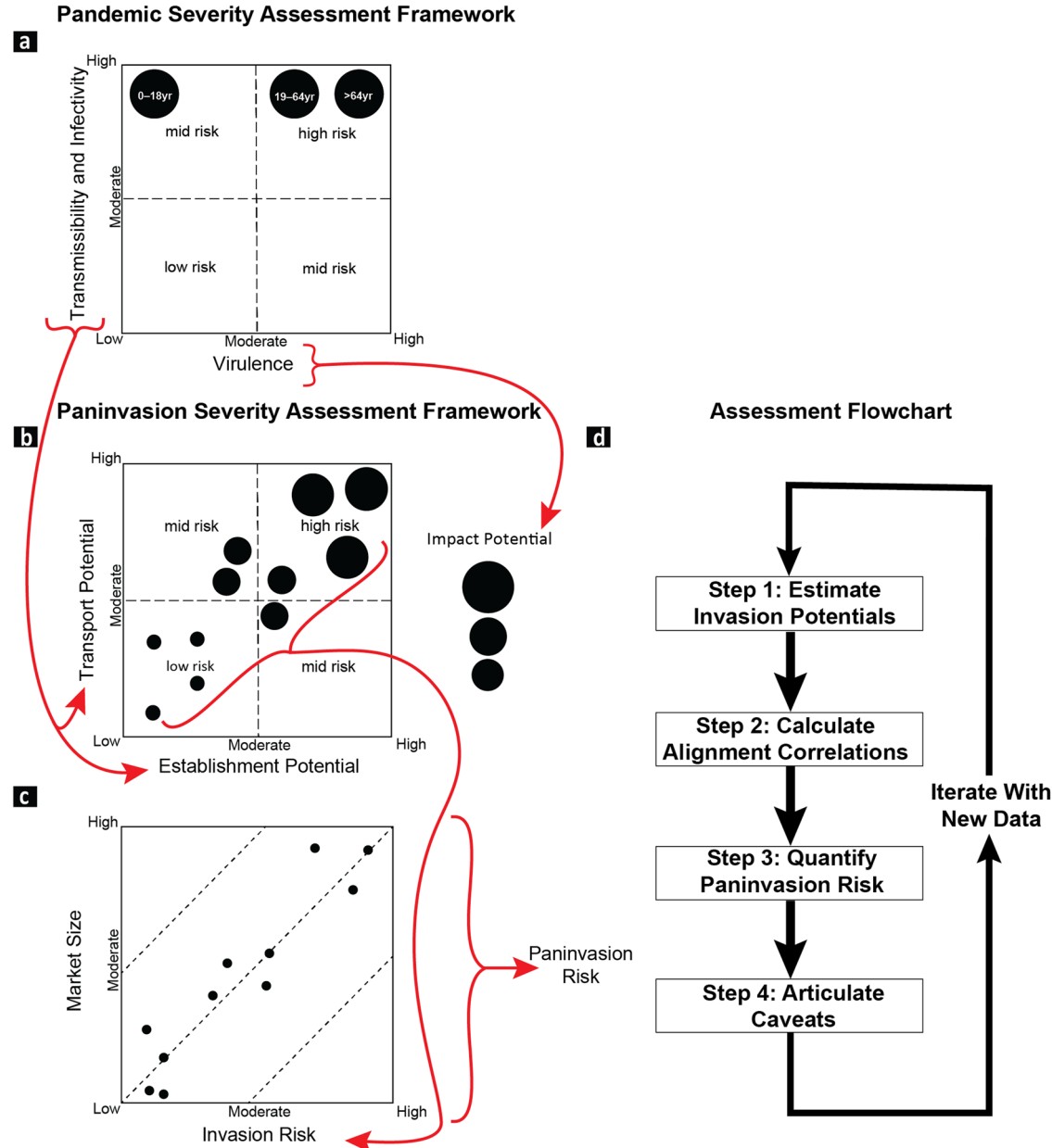

**Fig. 2 The paninvasion severity assessment framework is adapted from the U.S. CDC pandemic severity assessment framework used to estimate the risk of emerging human pathogens.** For pandemics (**a**), quadrant plots of pathogen transmissibility and infectivity (combined on one axis) vs. pathogen virulence (clinical severity) are used to compare the risk of a pathogen across different populations or age groups[8,9,11,72]. For paninvasions (**b**), invasion potentials for an emerging regional invasive species are estimated (**d** Step 1) by equating pathogen transmission with transport potential, infectivity with establishment potential, and virulence with impact potential (follow the red arrows) across regions (black circles) to construct quadrant plots that depict their alignment based on multivariate correlations (**d** Step 2; see Methods). Next, paninvasion risk (**c**) is estimated from the correlation between regional invasion risk estimated from the multivariate regression of invasion potentials (**d** Step 3; see Methods) and the size of regional markets that could be disrupted. The steps of the paninvasive severity assessment framework (**d**) culminate in articulated caveats in the current assessment that direct future research (**d** Step 4) that provides data to inform the next assessment iteration.

## Results

**Spotted lanternfly invaded range**. To assess SLF paninvasion risk, we first estimated the current U.S. invaded range (ca. 2020). We aggregated distributional data from multiple sources, including announcements made by state departments of agriculture on cargo interceptions that did not lead to established populations (i.e., regulatory incidents). By 2020, SLF had been established in nine U.S. states, with clear incidents of long-distance transport and establishment in Virginia, New York, and Pennsylvania. In eight additional states, individuals were intercepted in cargo and on transported goods originating from states with established SLF. California had intercepted the most with >40 individual SLF on 35 flights found during cargo inspections, but all were dead or moribund and not egg masses[38]. No international reports of regulatory incidents from the U.S. have been published. These regulatory incidents suggest that cargos with SLF were frequently transported from the invaded range in the U.S. northeast to at least as far as to the U.S. west coast (Fig. 3).

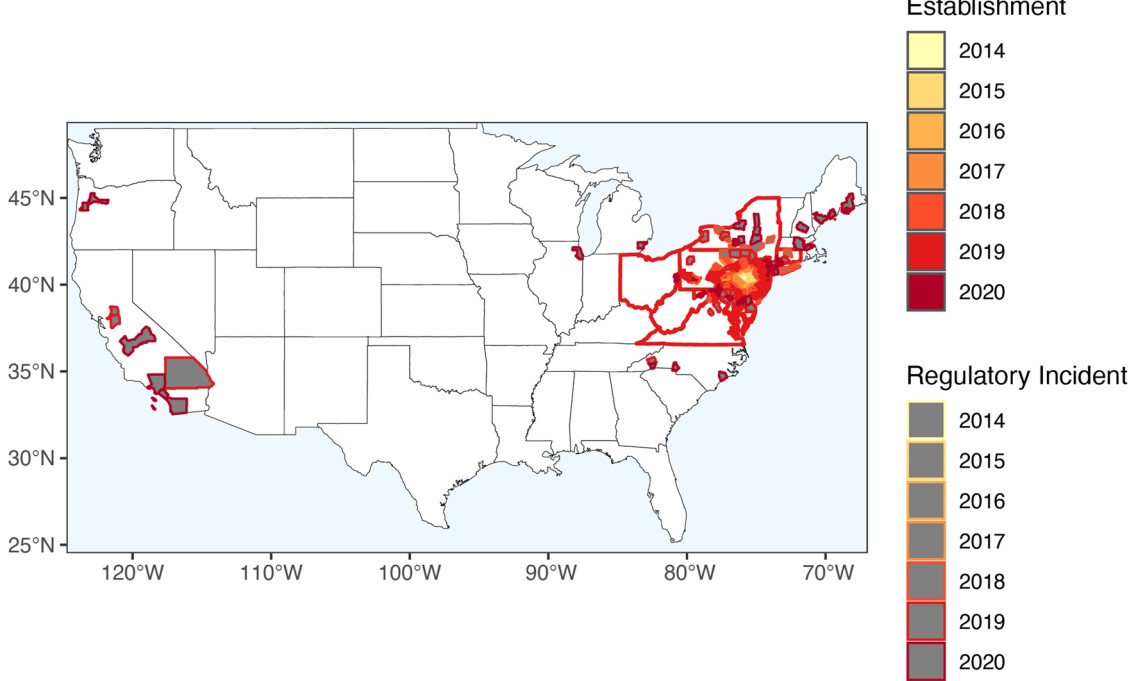

**Fig. 3 Spotted lanternfly (*Lycorma delicatula*, SLF) invaded nine states in the U.S. by 2020.** Colored polygons are counties with established SLF populations. Gray-filled polygons are counties where SLF have been transported but have not been established. These regulatory incidents include any egg cases, living, moribund, and dead individuals found in cargo. Red outlined states have established SLF populations.

**Alignment of invasion potentials**. We estimated the three invasion potentials—transport, establishment, impact—for 50 U.S. states and 223 countries. For transport potential, we used the total metric tonnage of goods imported from invaded states[39,40]. The current SLF spread in the U.S. (Fig. 3) was explained by total tonnage (Supplementary Methods, Supplementary Table 1). States with the highest transport potential were mostly in the eastern U.S., but Illinois, Texas, and California also heavily traded with the invaded states, indicating that SLF had a high potential to be transported both regionally and transcontinentally (Figs. 4a and 5a). Globally, transport potentials were highest in several European countries, Canada, and Brazil (Figs. 4b and 5b).

We based establishment potential on an ensemble estimate of SDMs built on SLF and TOH geolocations (see Supplementary Methods for SDM methods, Fig. 4)[41]. Our ensemble estimate of SLF establishment potential was spatially similar to other SDMs[42,43] and physiologically based demographic models of SLF[44,45]. However, our estimate indicated urban landscapes as likely establishment locations and showed fine spatial-scale variation in establishment potential (see our interactive map https://ieco-lab.github.io/slfrsk/). For each state and country, we extracted the mean, median, and max predicted suitability to estimate establishment potential (Supplementary Methods, max is presented in Fig. 5). Most U.S. states had high establishment potential (Figs. 4a and 5a), and all the countries with the highest transport potential also had the highest establishment potential (Figs. 4b and 5b).

We estimated SLF impact potential as the annual average tonnage of grapes and wine produced for each U.S. state and country[46–48]. States and countries with many important viticultural regions were geographically clustered (Fig. 4). Those with the highest impact potential included: California, Washington, and Oregon on the U.S. west coast; New York and Pennsylvania on the U.S. east coast (Figs. 4a and 5a); and Italy, France, and Spain in western Europe (Figs. 4b and 5b). After the discovery of the Pennsylvania invasion in 2014, it only took a few years for neighboring states to be invaded as well (Fig. 3). Thus, should one region on the U.S. west coast or in western Europe become invaded by SLF, neighboring states and countries are likely to also become invaded quickly.

SLF invasion potentials across states and countries were aligned (Fig. 5). Alignment correlations calculated as Spearman's rank correlation ($\rho$ statistic) among transport, establishment, and impact potentials were positive for impact potential measured as state grape production ($\rho = 0.41$, $P < 0.005$, Fig. 5a), state wine production ($\rho = 0.52$, $P < 0.001$), country grape production ($\rho = 0.67$, $P < 0.001$, Fig. 5b), and country wine production ($\rho = 0.63$, $P < 0.001$). This alignment of potentials is clear in the invasion potential alignment plots (Fig. 5). Major grape-producing regions fall in the upper-right quadrant of the plots where regions have both high transport and high establishment potentials.

**Paninvasion risk**. We estimated the risk of SLF to disrupt the global wine market to be an 8 out of 10 (Fig. 6). To derive this value, we regressed country grape production on country transport and establishment potentials. Each predicted value from this multivariate regression can be considered an estimate of the risk of SLF to invade and impact a country's grape production. We then rescaled these predicted values from 1 to 10 and correlated them to wine export market size ($\rho = 0.66$, $P < 0.001$). To place SLF on a scale of paninvasion severity, we rescaled the correlation coefficient, $\rho$, from 1 to 10, so that 1 is a complete negative correlation and 10 is a complete positive correlation between predicted impact and market size. Low values on this scale indicate that the global market is buffered against a paninvasion, while high values indicate that a paninvasion is likely unless mitigation actions are taken to reduce invasion potentials.

**Discussion**
The risk of an SLF paninvasion is high and coordinated effort should be made to reduce its transport, establishment, and impact

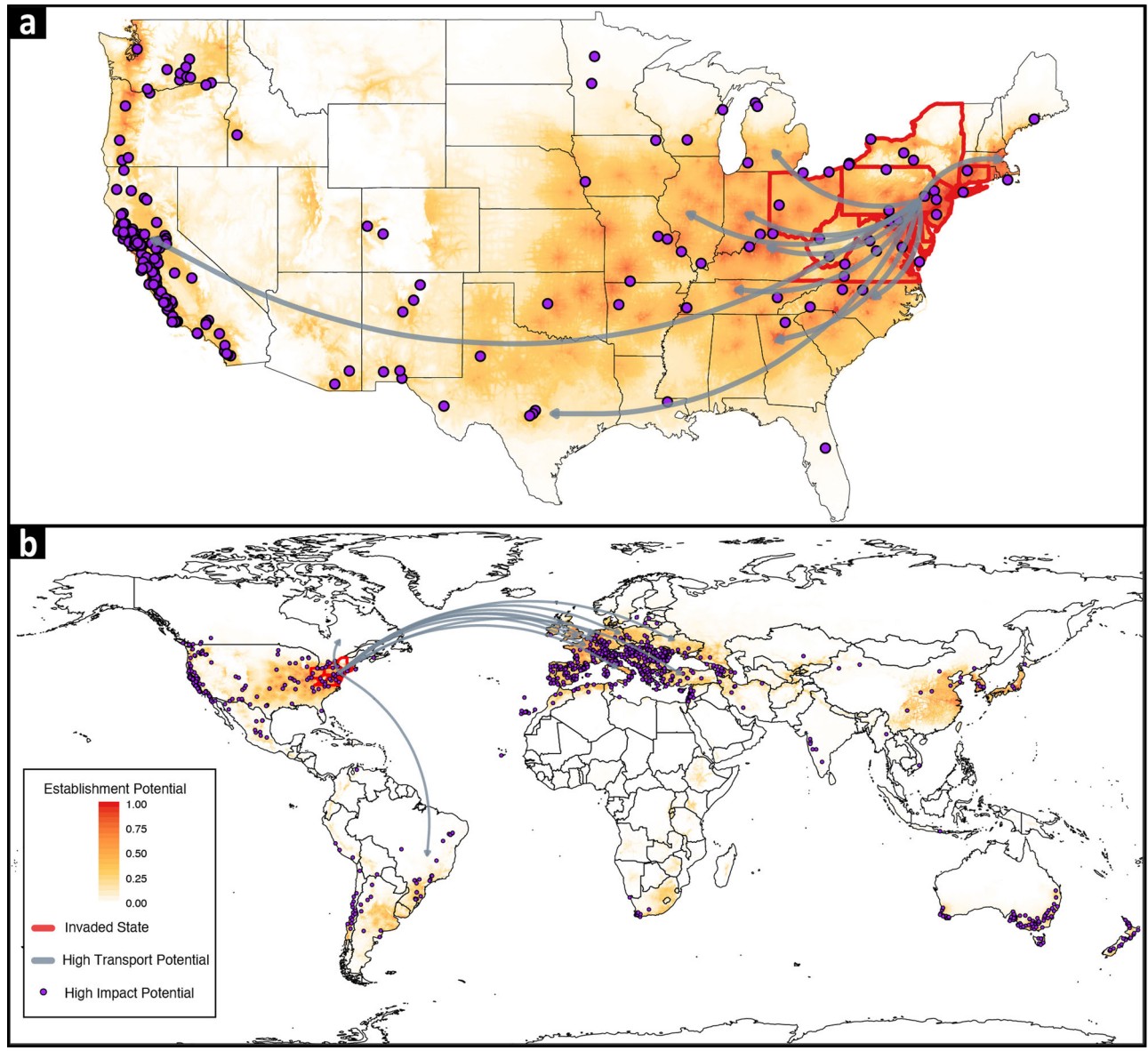

**Fig. 4 Strong spatial alignment of spotted lanternfly (*Lycorma delicatula*) transport, establishment, and impact potentials exists.** Arrows point from the U.S. invaded region to the top ten states (**a**) and countries (**b**) with the highest transport potentials. Purple points are locations of important viticultural areas. Map shading is an ensemble of species distribution models that estimates establishment potential.

potentials globally. In the U.S., efforts to reduce SLF transport potential are primarily through quarantine and inspection of goods. SLF is a regulated plant pest and the U.S. Department of Agriculture (USDA) is working toward implementing consistent, science-based, and nation-wide transport protocols[21,49,50]. We recommend that estimates of SLF transport potential be updated regularly as more states become invaded. Finer spatial and temporal scale data are needed on high transport potential pathways such as rail, landscaping stone, and live tree shipments to better forecast long-distance transport[18,21,34]. Egg masses in diapause are the most likely life stage to result in new satellite populations if transported. Thus, models should integrate annual trade dynamics with SLF annual phenology[44,51]. Such models would identify goods to quarantine and inspect that are transported out of the invaded range when eggs are in diapause before spring hatch[52].

The main host plant for SLF is the globally invasive TOH and reduction of establishment potential in the U.S. focuses on removing and treating TOH with herbicide[18]. The USDA and state departments of agriculture are the main agencies tasked with SLF control. However, these agencies lack the resources to manage TOH at both broad and local scales needed to reduce establishment potential, and businesses and private citizens are increasingly burdened with TOH removal costs[21]. Future research should focus on cost-effective, targeted TOH biocontrol methods that can be implemented at broad spatial scales[53]. We suggest increased public funding for local TOH management around vineyards and properties at risk of transporting SLF[21]. Currently, TOH seeds are inexpensive and easily purchased online. We suggest eliminating the horticultural sale of TOH globally. The U.S. federal and state governments should put TOH on their noxious weed lists, making it unlawful to sell, grow, or move this weedy invasive[54]. As SLF spreads, it will encounter new hosts besides TOH. To anticipate novel hosts, association studies that track SLF survivorship, development speed, and attraction are needed across a wide range of potential hosts, especially for those in landscapes surrounding high transport and impact potential locations like railyards and vineyards[22,30,31,55].

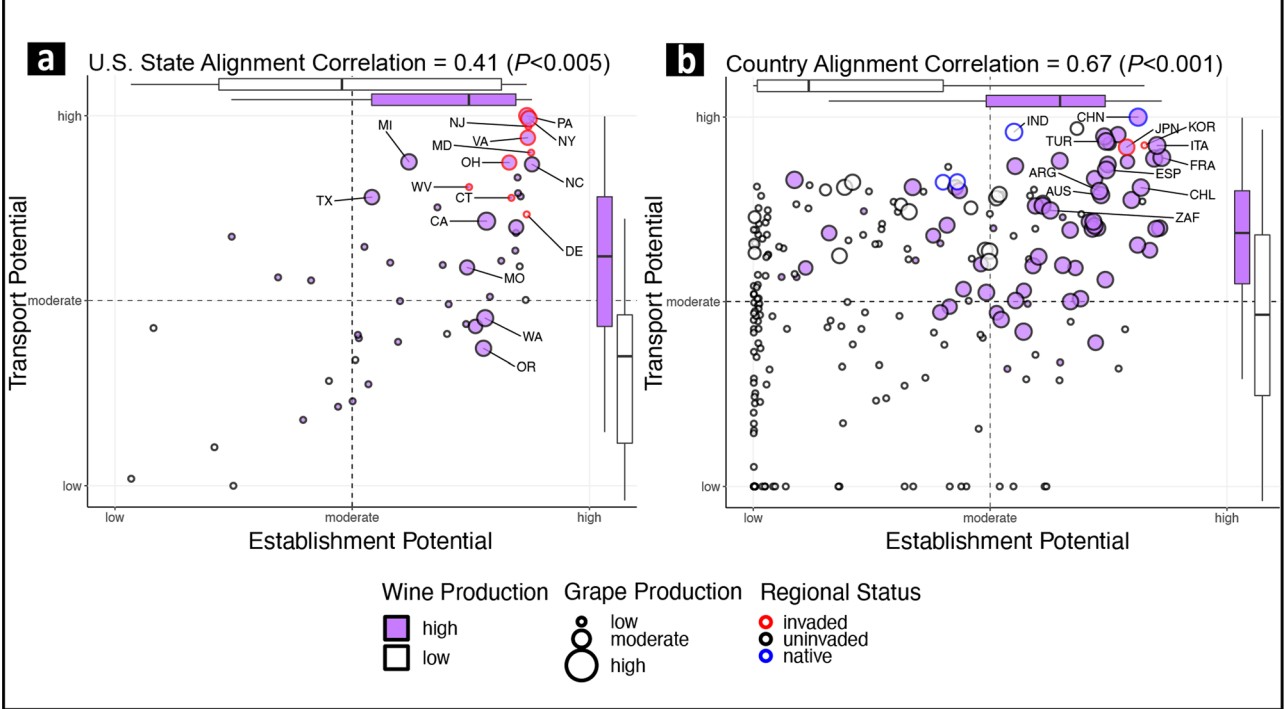

**Fig. 5 Spotted lanternfly (*Lycorma delicatula*) invasion potentials are aligned for wine and grape-producing regions.** States (**a**) and countries (**b**) that produce most of the global supply of grapes (point size) and wine (point fill color) also have high transport and establishment potentials. Invaded regions and the top-10 grape-producing regions are labeled; box plots split the distributions of establishment potential and transport potential into high (purple) and low (white) wine-producing regions (center line is the median, box limits are the upper and lower quartiles, and whiskers are 1.5 × the interquartile range); dashed lines divide the data into high-high, high-low, and low-low potential quadrants; and axes are scaled and formatted as suggested by the pandemic severity assessment framework[8,9]. Point color indicates high and low wine-producing regions. Point size depicts grape production in metric tons.

Reduction to SLF impact potential currently relies on reducing populations with tree-band trapping and broad-spectrum insecticides (e.g., carbamates, organophosphates, pyrethroids, neonicotinoids) that have high nontarget mortality[18,23,56,57]. However, existing management practices do not prevent vineyard reinfestations and pesticide application often overlaps with grape harvest when adults move into vineyards[36]. Damaged vines can be pruned but grape yield is reduced[58]. More research on long-term control methods for established populations is critical for reducing SLF impact potential. First, trapping technologies that reduce bycatch must be refined and widely deployed[57,59]. Second, biocontrol agents that specialize in SLF, such as parasitoid wasps and fungus show promise, but more work is needed to understand nontarget attack rates[60–62]. Finally, like targeted mRNA vaccines developed to reduce the impact of SARS-CoV-2 on human health[63], SLF-specific RNAi insecticides have the ability to control outbreaks in vineyards and beyond[21,64]. Although SLF impacts on vineyards within its invaded range are significant, to date, SLF has yet to invade a major viticultural area. Its actual impact on such regions with larger, wealthier, and interconnected wine economies is thus unknown. It is also unclear whether market elasticity might weaken or strengthen the disruption of an SLF paninvasion to the global wine market. It behooves governments to heed the paninvasion risk of this grapevine pest.

When a pest like SLF with high paninvasion risk emerges, coordinated governmental efforts can mitigate global market disruptions. For example, the Great Wine Blight of the late nineteenth century caused by grapevine phylloxera (Hemiptera: *Daktulosphaira vitifoliae*) was the largest shock to the global wine market ever recorded[65]. Phylloxera decimated European vineyards, but the market recovered due to pest management solutions whose development was coordinated by high-level officials in the French federal government[65].

For SLF in the U.S., early federal coordination was hampered. The U.S. National Invasive Species Council (NISC) contains high-level federal officials (e.g., Secretaries of State, Interior, Agriculture, and Defense) who coordinate the reduction of invasive species impact[66]. In 2019, NISC funding was cut and the Invasive Species Advisory Committee (ISAC) was dissolved. The ISAC comprised scientific experts who advised the NISC by producing memoranda and white papers on emerging invasive species issues[67]. These actions decreased U.S. capacity to respond to emerging paninvasive species[68]. In 2021, the ISAC was reinstated, and NISC's funding is expected to be fully restored[69]. Based on our SLF risk assessment, we suggest the reinstated ISAC produce memoranda and white papers on solutions for SLF as soon as possible. SLF risk must be communicated to the NISC, who can mobilize the resources needed to reduce its impact.

The paninvasion severity assessment framework is a stakeholder communication tool to assess if an invasive can cause market, environmental, and human-health disruptions on a global scale (Fig. 1). It does so by equating pathogen transmission, infectivity, and virulence—well known to the public due to COVID-19 pandemic—with invasive species transport, establishment, and impact potentials (Fig. 2). Paninvasion assessments produce accessible maps (Figs. 3 and 4), scatter plots (Fig. 5), and easy to understand risk values (Fig. 6). Going forward, invasion potentials for other species are likely to increasingly align and coordinated governmental efforts will be needed to reduce such potentials in the U.S. and internationally. It is prudent that when any new invasive species is found, the severity of its paninvasion risk be assessed.

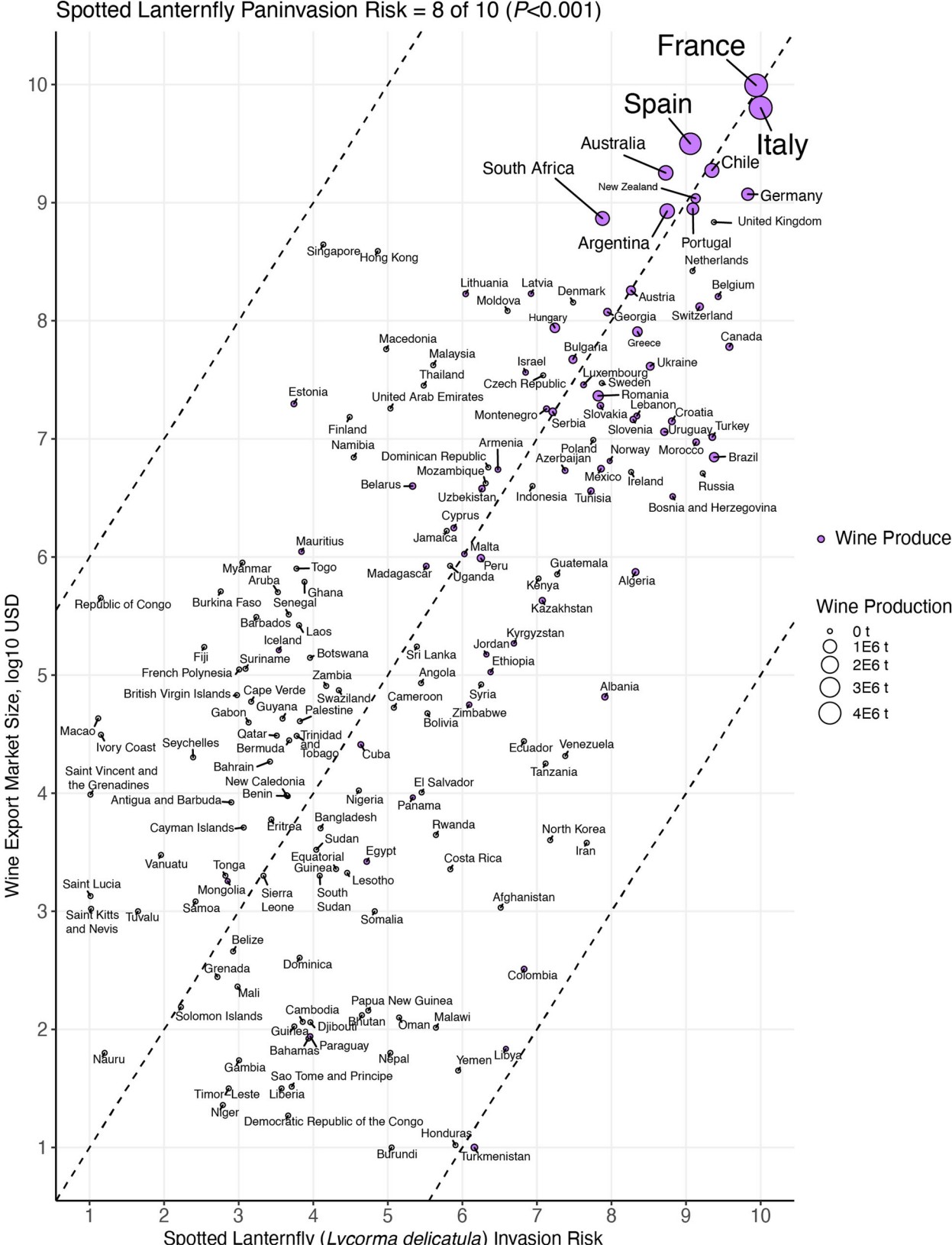

**Fig. 6 Spotted lanternfly (*Lycorma delicatula*) paninvasion risk is an 8 out of 10 due to the correlation between country wine export market size and invasion risk.** Dashed lines are one-to-one guidelines at −0.5, 0, 0.5 intercepts. Point color indicates high and low wine-producing regions. Point and label size depict wine production in metric tons.

## Methods

Here, we provide a methodological discussion of the paninvasion severity assessment framework (Fig. 2) and its application to SLF (Fig. 1). To make the SLF assessment easy to refine once new data and insights are available, we provide both an open-source R package that includes all data to reproduce all results (https://ieco-lab.github.io/slfrsk/) and a Google Earth Engine application to map SLF paninvasion severity from global to local scales (https://ieco-lab.github.io/slfrsk/articles/vignette-040-ee-data.html). These open-science tools also are adaptable to other emerging regional invasives at risk of paninvasion, e.g., [70]. Here, we focused on agricultural and economic data most relevant to assess pest risk, but for non-pest invasives, data on environmental and human health may be a higher priority.

**Paninvasion severity assessment framework**. Although the invasion process can be divided into many stages, the paninvasion severity assessment framework focuses on the three main stages most often estimated in invasion risk assessments[2] and that is analogous to the disease potentials that public-health scientists quantify for pathogens (Fig. 2)[8]. When a pathogen with pandemic risk emerges, public-health scientists place it within scaled measures of transmissibility and infectivity (often combined and termed transmissibility), and virulence (clinical severity) to assess its risk[8,9]. For example, when SARS-CoV-2 emerged during the COVID-19 pandemic, the initial understanding was that different age groups had similar potentials to transmit and become infected (Fig. 2a, y-axis), but different age groups varied in their clinical severity once infected (Fig. 2a, x-axis)[8,71,72]. To adapt this public-health framework to invasion process theory[2,73–76], we equated transmission, infectivity, and virulence potentials of a pathogen across different human populations to the transport, establishment, and impact potentials of an invasive species across different regions (see colored arrows between Fig. 2a, b). For example, in Fig. 2b we placed several hypothetical regions that together indicate strong alignment (i.e., multivariate correlation) among invasion potentials across the regions. In this example, predicted invasion risk (Fig. 2c, x-axis) for these three hypothetical regions is strongly correlated to a measure of their contributions to a global market (Fig. 2c, y-axis), indicating an overall high paninvasion risk.

Paninvasion assessments comprise four steps (Fig. 2d): (1) estimate invasion potentials, (2) calculate alignment of invasion potentials, (3) quantify paninvasion risk, and (4) articulate caveats, which we describe in detail for SLF below and in the Supplementary Methods.

**Step 1: estimate invasion potentials**

*Transport potential*. Transport potential is a measure of propagule pressure[77]. The prevailing hypothesis on SLF transport potential is that regions that import more tonnage of commodities from the invaded U.S. region also import more total tonnage of goods and trade infrastructure (e.g., cargo containers, pallets, and railcars) that inadvertently transport SLF propagules, such as egg masses, long-distances[18,21,26,34,78,79]. To estimate which states were invaded and identify SLF transportation events, we obtained a database of SLF records from the USDA and aggregated first-find and regulatory incident reports, e.g., [38]. As of December 2020, the invaded states were Connecticut, Delaware, Maryland, New Jersey, New York, Ohio, Pennsylvania, Virginia, and West Virginia (Fig. 3). We estimated transport potentials from the U.S. invaded region as the $\log_{10}$ of the average annual metric total tonnage of all goods imported between 2012 and 2017 by states and countries from the invaded U.S. states. This date range encapsulates both pre- and post-introduction of SLF to the U.S. and maximized temporal overlap across different data sources. Tonnage data from these invaded states were from the U.S. Freight Analysis Framework for interstate imports[39] and from the U.S. Trade Online database for international imports[40], both accessed on June 14, 2019. Current SLF spread in the U.S. was explained by our transport potential metric suggesting that using total tonnage is a valid metric of transport potential for SLF (Supplementary Methods, Supplementary Table 1).

*Establishment potential*. Establishment potential is the set of species-specific and environmental characteristics of a region that determine if a transported species can generate a self-sustaining population[2]. We determined SLF establishment potential from an ensemble estimate from three global SDMs: a multivariate SDM of TOH (sdm_toh), a multivariate SDM of SLF (sdm_slf1), and a univariate SDM of SLF that modeled SLF presence on the predicted values from sdm_toh (sdm_slf2). Models were constructed using MaxEnt ver. 3.4.1 according to unbiased niche modeling best practices (see Supplementary Methods)[80–82]. Specifically, our SDMs were built from unique, error checked, and spatially rarefied presence records: sdm_toh on 8,022 TOH presence records, and sdm_slf1 and sdm_slf2 on 325 SLF presence records obtained from GBIF on October 20, 2020. To find the best-fit models that explained TOH and SLF presences, we identified a subset of six covariates, from 22 candidate covariates[83–86], that minimized model collinearity: annual mean temperature, mean diurnal temperature range, annual precipitation, precipitation seasonality, elevation, and access to cities. We fit sdm_toh and sdm_slf1 with these six covariates. sdm_toh represents our best estimate of the global distribution of TOH, thus we fit sdm_slf2 that modeled SLF suitability from the predicted values of sdm_toh. As such, sdm_slf2 represents suitability that considers a primary plant host (TOH)[25] that is also invasive but likely not at equilibrium[87] and the same abiotic covariates as sdm_slf1 (sdm_toh uses the same covariates). We evaluated model performance with k-fold cross-validation,

specifically the receiver operating characteristic of the area under the curve and omission error[88–90].

Each model estimated suitability at a 30-arcsecond (at the equator approximately 1 km²) global resolution with pixel values scaled 0–1, which we averaged across models per pixel to produce one ensemble image and intersected this image with state and country polygons[82]. Establishment potential for the 50 U.S. states and 223 countries was estimated as the maximum pixel value for each state and country. Results and conclusions with mean and median pixel values instead of max were similar (see https://ieco-lab.github.io/slfrsk/).

*Impact potential*. We used $\log_{10}$-transformed average annual production tonnages of grapes and wine as two separate estimates of impact potential. For consistency, we used grape and wine production during the same span of time as transport potential estimates, 2012–2017. Grape production for countries was from the Food and Agriculture Organization of the United Nations crop database[46] and for states from the USDA National Agricultural and Statistics Service commodity database[47], both accessed on January 24, 2020. Wine production in metric tons was from FAOSTAT for countries[46], accessed on June 21, 2019, and from the Alcohol and Tobacco Tax and Trade Bureau (TTB) for states[48], accessed on June 22, 2019, which was in gallons but we converted it to metric tons assuming 3.776e-3 t/gallon[91]. Major viticultural regions (Fig. 4) were aggregated and georeferenced from a TTB U.S. state data set[92] and the global viticultural regions Wikipedia list[93] to better visualize impact within states and countries, both accessed on April 22, 2020.

**Step 2: calculate alignment correlations**. To consider how all three invasion potentials may coincide for regions, we calculated alignment correlations for states and countries separately. Alignment was calculated for each of the two measures of impact potential as Spearman rank correlations between impact potential and the predicted values from linear regression models of each impact potential regressed on transport and establishment potentials together[94]. We then visualized these multiple multivariate correlations as quadrant plots following a stakeholder-friendly and approachable format adapted from the pandemic severity assessment framework[8,9].

**Step 3: quantify paninvasion risk**. To determine if invasion risk for countries corresponds with economic impact on the global wine industry, we investigated the relationship between wine market size and predicted risk of invasion for individual countries. We estimated wine market size for 223 countries (including some that export but do not produce wine) as the value of wine exports corresponding with the years for our trade data (2012–2017, $\log_{10}$ USD) downloaded from the FAOSTAT detailed trade matrix[46], accessed August 31, 2020. Then, we regressed country grape production on transport and establishment potentials with multiple linear regression. Each predicted value from this regression can be considered an estimate of the risk of SLF to invade and impact a country's grape production. We rescaled these estimates from 1 to 10 to create an easily interpreted estimate of risk and then correlated these predicted values to wine export market size. To place overall SLF paninvasion severity on a clear scale for both researchers and stakeholders, we simply rescaled the Pearson correlation from 1 to 10, so that 1 is a complete negative correlation and 10 is a complete positive correlation between country risk and wine export market size.

**Step 4: articulate caveats**. Paninvasion risk assessments should be performed iteratively as the invasion process continues across regions and responses are mobilized. Early assessments of emergent pests have great utility to support early responses but often come with caveats. The fourth step of the framework is to articulate the caveats of a current assessment to guide future research. These caveats should explicitly consider the assumptions made when estimating invasion potentials and how those potentials and risk could change as the invasion process continues. Below we articulate caveats of our SLF assessment to provide a basis for future refined assessments of SLF to disrupt the global wine market.

We calculate paninvasion risk of SLF via stepping-stone transport from the eastern U.S. However, major wine-producing nations also heavily trade with China, Japan, and South Korea, where SLF is also established. Total SLF transport potential is thus greater than our estimates, meaning paninvasion risk is higher than what we report here. Future work should account for global trade network dynamics with other nations with established SLF populations. However, comprehensive surveys are first needed on the distribution of SLF in these other countries and the trade emanating from regions with established populations. Further research should focus on the identification of which industries and commodities are most likely to increase long-distance spread. This requires linking trade dynamic models to phenological models to indicate if high-risk trade is occurring at the same time as egg laying, because eggs are the life stage most likely to be transported long-distances[21]. In addition, refinement of SLF propagule pressure dynamics[77], specifically propagule number and ratio of successful to failed transportation events, can improve estimates of transport potential.

Establishment potential should be improved as additional data and models become available. Because we use MaxEnt, a presence only, correlative SDM method to estimate establishment, we measure suitability for SLF in a way that does

not account for how SLF population density and possible plastic and adaptive responses to novel environmental conditions in invaded regions may affect establishment success. Omission of population density and other demographic variables can hinder accurate prediction, especially for SDMs[95], and whenever possible, priority should be placed on using them alongside models that rely on pest physiology to predict establishment potential[96]. For SLF, two physiologically based models[44,45] largely correspond with our SDM-based establishment potentials and thus support the global establishment potential for SLF we report here. Early assessments of paninvasion severity are unlikely to account for plasticity and adaptation that is common for invasive pests, especially when combined with variation in local weather patterns and climate change. Indeed, a recent analysis suggests that SLF will experience increased suitable habitat and a greater impact in China in the future due to climate change[97]. The expected effect of climate change is likely more complicated for SLF, which has a flexible life cycle that can include but does not require, temperature-linked diapause for overwintering in cooler regions[24,98,99]. Survivorship appears greater without such diapause[98], and thus establishment potential may be even higher than expected in warmer climes.

Variation in weather, climate change, host preference, and pest density can influence pest impacts. Such factors often act at different scales and in a spatially heterogenous manner. For example, SLF prefers grapes, but the degree to which it does over alternative hosts near vineyards remains poorly known, which is important, since SLF appear to have their highest densities at vineyard edge habitats[100]. The vulnerability of viticultural regions may be affected by the prevalence of particular grape cultivars or alternative hosts, but additional research is necessary to elucidate SLF feeding preference. Similarly, the relationship between SLF density within a vineyard and in the surrounding landscape remains poorly known, which is in turn likely to be influenced by weather patterns and plant phenology[36]. As SLF host preference and its relationship to landscape variables become better understood, they should be incorporated into considerations of impact potential.

Lastly, to refine the SLF paninvasion risk assessment, future work should calculate invasion potentials for other grape pests like phylloxera to place SLF on an absolute scale of risk severity[101]. Our assessment of SLF relativizes invasion potentials with the assumption that regions with high potentials relative to other regions also have high absolute potentials. SLF has broad environmental suitability, a flexible life cycle, ability to lay many discrete egg masses on numerous substrates (Fig. 1b), observed rapid spread (Fig. 3), and realized impacts on grape and wine production[21–26,36], so its absolute potentials are likely very high. However, absolute potentials can only be assessed by comparing multiple paninvasive species, like what is done for pathogens. When a pathogen with pandemic potential emerges, the pandemic severity assessment framework compares the severity of the potentials of the current outbreak pathogen to past pandemic-producing pathogens[8,9,71]. The next step toward a mature paninvasion framework is to estimate invasion potentials for current paninvasive species, so that the likelihood of a paninvasion for any emerging regional pest can be placed on an absolute scale of severity.

**Reporting summary**. Further information on research design is available in the Nature Research Reporting Summary linked to this article.

## Data availability

All datasets generated during and/or analyzed in this study are available as described in the methods and research compendium (https://ieco-lab.github.io/slfrsk) or can be obtained as a part of the companion R package (https://github.com/ieco-lab/slfrsk). Large data files (e.g., SDM files) are available in a Data Dryad repository (https://doi.org/10.5061/dryad.msbcc2g1b).

## Code availability

All codes written and used for this study are available as an R package that is publicly available at: https://github.com/ieco-lab/slfrsk.

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

## Acknowledgements

We thank Julie Urban, Heather Leach, Nadège Bélouard, Stefani Cannon, Seba De Bona, Jason Gleditsch, Stephanie Lewkiewicz, Victoria Ramirez, Payton Phillips, Timothy Swartz, and the Integrative Ecology Lab at Temple University for comments and feedback on earlier drafts of this manuscript. This work was funded by the United States Department of Agriculture Animal and Plant Health Inspection Service Plant Protection and Quarantine under Cooperative Agreements AP19PPQS&T00C251 and AP20PPQS&T00C136; the United States Department of Agriculture National Institute of Food and Agriculture Specialty Crop Research Initiative Coordinated Agricultural Project Award 2019-51181-30014; and the Pennsylvania Department of Agriculture under agreements 44176768, 44187342, and C9400000036.

## Author contributions

All authors conceived and designed the study. N.A.H. acquired and analyzed data. All authors contributed materials/analysis tools and wrote the paper.

## Competing interests

The authors declare no competing interests.
