## [Peer Review File · Communications Biology]

Reviewers' comments:

Reviewer #1 (Remarks to the Author):

Leveraging an increasingly universal approach to risk assessment in the post-COVID-19 era, Huron and colleagues adapt the public-health US CDC pandemic severity assessment framework to the assessment of risk posed by global pest invasions that the authors call paninvasions. The authors apply this new framework to the spotted lanternfly planthopper (*Lycorma delicatula*; SLF) that invaded the US in circa 2014 on goods imported from its native range in China. Because SLF feeds on grape, there is scope to assess its global invasive potential in uninvaded viticultural regions globally. Assessing the risk of SLF is important and of broad interest, also because this pest feeds on many other plant species including its preferred host, the cosmopolitan invader tree-of-heaven (*Ailanthus altissima*, TOH).

The paper states that "SLF greatly impacts grape production" but there seem to be no specific reference supporting this claim. For example, Urban (2020) points out that "[t]he ultimate impacts of this insect are not yet known". In this paper, the risk of SLF to the global wine industry is linked to the transport potential (e.g., global trade) of SLF to regions where grape is grown, to SLF establishment potential (MaxEnt SDMs), and to SLF impact potential estimated as the annual average production of grape and wine in a state or country. This method for assessing "paninvasion severity" conservatively assumes that the impact of SLF, once established, will be the same across its invaded range. However, severity will probably also depend on local weather patterns that may vary greatly across invaded locations and as a result of climate change.

The authors present their paninvasion severity assessment framework as a simple tool to assess invasion potentials for any emerging invasive species. While it is true that known pests (e.g., SLF) of crops that are grown globally (e.g., grape) should be analyzed as potential invasive pests, this requires the ability to predict accurately their potential geographic range and relative abundance in novel areas. Such accurate predictions may be unachievable using de facto standard correlative methods (e.g., MaxEnt) as recently shown for the South American tomato pinworm *Tuta absoluta* (Ponti et al. 2021, doi:10.1007/s10530-021-02613-5). Correlative methods such as MaxEnt and CLIMEX rely on occurrence records to assess invasive potential. However, suitable records capturing the full range of plasticity to environmental variables may not be available until after the potential invasive range of an invasive species has been occupied. This should probably be listed as an additional caveat of the present study.

Another point that could be clarified is why the `sdm_slf2` based on the TOH SDM was fitted and what is the added value of `sdm_slf2` to the analysis.

Reviewer #2 (Remarks to the Author):

The paper attempts to adapt a public-health framework to invasion science (invasive species) "leveraging an increasingly universal risk vocabulary of pandemics" of infective agents such as covid (organism to organism contact) to the spread and impact of invasive species (I suppose pests and weeds) coining the term paninvasion (pan invasion) to describe their spread, establishment, and impact. The authors presume that a framework for such analyses of invasive species does not exist, and yet for many decades it has been the underlying bases of the field of biological control (i.e., transport (by whatever mechanism be it natural or anthropogenic), establishment (is weather suitable and are hosts available?), and what is the impact potential. Specifically, they use as their foil for their paninvasion severity assessment framework the spreading of the regional invasive US grape pest, the spotted lanternfly planthopper (*Lycorma delicatula*; SLF), and its assumed potential to "disrupt the global wine" market. "Once established, SLF likely has high global impact potential to wine markets because grape is an equally suitable host", – i.e., but it is not equally preferred compared to its invasive host *Ailanthus altissima* (tree of heaven) also native to China. Using the SDM MaxEnt, they found the apparent future distribution of lantern fly is roughly same as that of its preferred host (*A. altissima*) used as

component to estimate establishment potential (i.e., worldwide) that in the palearctic is similar to that of grape. Using multivariate statistics (multiple linear regression?), "...We found that SLF invasion potentials are aligned globally because important viticultural regions with suitable environments for SLF establishment also heavily trade with invaded US states". They state the "Alignment correlations among transport, establishment, and impact potentials were positive for impact potential measured as state grape production ($\rho = 0.41$, $P < 0.001$), etc.", and yet the meaning of alignment correlations and statistic ρ were not defined. The language is clear, but the meaning is opaque. For example, "Going forward, paninvasion potentials for other species are likely to increasingly align and coordinated governmental efforts will be needed to reduce invasion potentials in the US and internationally".

In summary, the major strength is the MaxEnt analysis (potential for establishment) that has previously been projected by another author using MaxEnt, while the major weakness is the transport mechanism that while likely valid conceptually for SLF is at best a very rough measure, but such measures are likely not valid for other plant/animal species that disperse/invade in a variety of ways. Their two caveats to quantifying invasion severity for SLF render mute their global assessments: paraphrased, (1) they estimate invasion severity of SLF via a stepping-stone introduction from the eastern US, but (2) because major wine producing nations also heavily trade with China where SLF is native and Japan and South Korea where SLF is established, SLF transport potential is greater than their estimates, meaning that invasion severity is also likely higher than what we report here, and future work should account for global trade network dynamics.

As cast, the transference of the notions of pandemics potential contributes little to the substance of the paper. The paper needs to be rewritten with the focus on the methods and data with the summary term paninvasion introduced to characterize its potential for spread widely. Figures 5 and 6 are interesting and with clear explanation of methods would make for an interesting paper. Figure 6 includes data for countries that are not wine produces. Importantly, they suggest more research on biocontrol which is likely the only long-term solution. I hope my comments will be useful in redrafting the paper.

Review Response for Huron et al.: Paninvasion severity assessment of a US grape pest to disrupt the global wine market (COMMSBIO-21-2451-T)

Corresponding Author: **Nicholas A. Huron**

Integrative Ecology Lab, Center for Biodiversity, Department of Biology, Temple University, Philadelphia, PA 19122 USA.

nahuron@temple.edu

Dear Drs. Karniski and Desneux and reviewers,

Thank you for considering our manuscript and providing reviews, which we have completely addressed. Below are the comments from Reviewers 1 and 2, pasted directly from the initial decision letter with our responses in green.

We thank both reviewers for their helpful comments that improve our manuscript. Thank you for considering our revised manuscript.

Sincerely,

Nicholas Huron

Reviewers' comments:

Reviewer #1 (Remarks to the Author):

Leveraging an increasingly universal approach to risk assessment in the post-COVID-19 era, Huron and colleagues adapt the public-health US CDC pandemic severity assessment framework to the assessment of risk posed by global pest invasions that the authors call paninvasions. The authors apply this new framework to the spotted lanternfly planthopper (*Lycorma delicatula*; SLF) that invaded the US in circa 2014 on goods imported from its native range in China. Because SLF feeds on grape, there is scope to assess its global invasive potential in uninhabited viticultural regions globally. Assessing the risk of SLF is important and of broad interest, also because this pest feeds on many other plant species including its preferred host, the cosmopolitan invader tree-of-heaven (*Ailanthus altissima*, TOH). The paper states that "SLF greatly impacts grape production" but there seem to be no specific reference supporting this claim. For example, Urban (2020) points out that "[t]he ultimate impacts of this insect are not yet known".

This is a good point that warrants clarification. The quote that is mentioned here from Urban 2020, which we cited, is introduced near the end of the abstract to highlight the emergent nature of SLF's invasion in the US. Urban also says "The longer-term impacts of SLF remain speculative, but could be significant." and that, "SLF is causing severe damage to vineyards from feeding and is a significant nuisance pest".

Furthermore, more work that we now cite has been published that assesses impact to grapes and to other plant species. This work clearly indicates that SLF is impacting grape production in the US invaded region, has the potential to impact grape production in the US for the long-term, and globally has already caused grape damage elsewhere (e.g., South Korea). In our manuscript, we now clarify that the long-term impacts of this insect are not yet known, because they have not yet spread into a major wine growing region.

To address this comment, we added the following text to the introduction and included the other citations that discuss grape impact:

Lines 51–3: SLF greatly impacts grape production^{22–26} and has been presented to the public as one of the worst invasive species to establish in the US in a century^{27–29}, but its paninvasion risk has not been assessed¹⁹.

Lines 62–6: Once established, SLF likely has high global impact potential to wine markets because grape is an equally suitable host. SLF develops at similar rates when fed grape or TOH, and fecundity increases when fed a mixed diet of these two preferred hosts^{25,30–33}. Asian vineyard production is impacted by SLF infestations^{34,35}; and SLF-invaded US vineyards have reported vine deaths, >90% yield losses, and closure (Fig. 1g–j)^{21,23,36}.

Lines 153–6: We suggest eliminating the horticultural sale of TOH; increasing funding for TOH removal; research on cost-effective TOH biocontrol methods^{9,51}; and because SLF are generalists, more research to identify other suitable hosts found in the landscapes surrounding high transport and impact potential locations like railyards and vineyards^{21,30,52}.

Lines 171–4: To date, SLF has yet to invade a major viticultural area, so its impact on such regions with larger, wealthier, and interconnected wine economies is unknown. It is also unclear whether market elasticity might weaken or strengthen the disruption of a SLF paninvasion to the global wine market.

In this paper, the risk of SLF to the global wine industry is linked to the transport potential (e.g., global trade) of SLF to regions where grape is grown, to SLF establishment potential (MaxEnt SDMs), and to SLF impact potential estimated as the annual average production of grape and wine in a state or country. This method for assessing “paninvasion severity” conservatively assumes that the impact of SLF, once established, will be the same across its invaded range. However, severity will probably also depend on local weather patterns that may vary greatly across invaded locations and as a result of climate change.

This is an excellent point. We were conservative in our estimates of paninvasion severity for reasons that the reviewer outlined.

To address this point made by the reviewer, we have expanded the caveats section to discuss the issue of climate change and local weather patterns. We also point out that our map (Figure 4) and our companion app (<https://ieco.users.earthengine.app/view/ieco-slf-riskmap>) show considerable variation within invaded and uninvaded regions for current establishment potential, which are indicative of local-scale variation. We also discuss how global warming could cause establishment potential in higher latitudes to increase, allowing cooler regions to provide enough degree days for complete SLF development. Below are the edits that we have made to the text:

Lines 363–82: Early assessments of paninvasion severity are unlikely to account for plasticity and adaptation that is common for invasive pests, especially when combined with variation in local weather patterns and climate change. Indeed, a recent analysis suggests that SLF will experience increased suitable habitat and a greater impact in China in the future due to climate change⁸⁹. The expected effect of climate change is likely more complicated for SLF, which has a flexible life cycle that can include but does not require, temperature-linked diapause for overwintering in cooler regions^{24,90,91}. Survivorship appears greater without such diapause⁹⁰, and thus establishment potential may be even higher than expected in warmer climates.

Variation in weather, climate change, host preference, and pest density can influence pest impacts. Such factors often act at different scales and in a spatially heterogeneous manner. For example, SLF prefers grapes, but the degree to which it does over alternative hosts near vineyards remains poorly known, which is important, since SLF appear to have their highest densities at vineyard edge habitats⁹². The vulnerability of viticultural regions may be affected by the prevalence of particular grape cultivars or alternative hosts, but additional research is necessary to elucidate SLF feeding presence. Similarly, the relationship of SLF density within a vineyard and density in the surrounding landscape remains poorly known, which is in turn likely to be influenced by weather patterns and plant phenology³⁶. As SLF host preference and its relationship to landscape variables become better understood, they should be incorporated into considerations of impact potential.

The authors present their paninvasion severity assessment framework as a simple tool to assess invasion potentials for any emerging invasive species. While it is true that known pests (e.g., SLF) of crops that are grown globally (e.g., grape) should be analyzed as potential invasive pests, this requires the ability to predict accurately their potential geographic range and relative abundance in novel areas. Such accurate predictions may be unachievable using de facto standard correlative methods (e.g., MaxEnt) as recently shown for the South American tomato pinworm *Tuta absoluta* (Ponti et al. 2021, doi:10.1007/s10530-021-02613-5).

This is a great point. We now include a discussion about the accuracy of predictions in our caveats section and cite this paper that developed a physiologically based demographic model for *Tuta absoluta*. New research on SLF also provides two lines of evidence that indicate that the establishment maps we produced are indicative of the potential geographic range and abundance of the pest.

First, two physiologically based demographic models of geographic range have been produced and both correspond strongly to our maps of establishment potential. These two models were built using different data and methods from each other and from our work. Combined, they provide strong support that SLF has global establishment potential. Further, they indicate areas of high SLF establishment potential in our estimates are also areas where SLF abundances are likely to be high.

Second, the geographic range of SLF (and many other pest insects) is influenced by diapause. Pests that undergo diapause can overwinter and thus expand into colder climates than pests that have not evolved diapause. SLF is a pest that is plastic in diapause—it undergoes diapause in cooler climates but loses diapause in warmer climates. In our caveats and supplementary methods, we discuss how SLF’s plastic diapause response increases

establishment potential. Specifically, we note that our modeled establishment potential is likely conservative estimate, especially for warmer regions based on higher survivorship of SLF that do not undergo chilling diapause. We have incorporated these considerations in our results and into our caveats section as follows:

Lines 99–104: Our ensemble estimate of SLF establishment potential was spatially similar to other SDMs^{42,43} and physiologically based demographic models of SLF^{44,45}. However, our estimate indicated urban landscapes as likely establishment locations and showed fine spatial-scale variation in establishment potential (see our interactive map <https://ieco.users.earthengine.app/view/ieco-slf-riskmap>).

Lines 354–63: Establishment potential should be improved as additional data and models become available. Because we use MaxEnt, a presence only, correlative SDM method to estimate establishment, we measure suitability for SLF in a way that does not account for how SLF population density and possible plastic and adaptive responses to novel environmental conditions in invaded regions may affect establishment success. Omission of population density and other demographic variables can hinder accurate prediction, especially for SDMs⁸⁷, and whenever possible, priority should be placed on using them alongside models that rely on pest physiology to predict establishment potential⁸⁸. For SLF, two physiologically based models^{44,45} largely correspond with our SDM-based establishment potentials and thus support the global establishment potential for SLF we report here.

Correlative methods such as MaxEnt and CLIMEX rely on occurrence records to assess invasive potential. However, suitable records capturing the full range of plasticity to environmental variables may not be available until after the potential invasive range of an invasive species has been occupied. This should probably be listed as an additional caveat of the present study.

We now include this as a caveat in our study as we stated above.

Another point that could be clarified is why the *sdm_slf2* based on the TOH SDM was fitted and what is the added value of *sdm_slf2* to the analysis.

We clarify this point in the manuscript. Specifically, the use of *sdm_toh* as an environmental layer for *sdm_slf2* takes into consideration that it is unlikely that TOH has reached full saturation of its distribution globally but that it is an important factor for SLF establishment potential. Given that we present an ensemble of the establishment models for our analyses, *sdm_slf2* provides a more specific consideration of TOH as an important plant host. A full description of modeling methods is included in the Supplementary Information. To clarify the use of this model, we change the following methods passage:

Lines 265–70: We fit *sdm_toh* and *sdm_slf1* with these six covariates. *sdm_toh* represents our best estimate of the global distribution of TOH, thus we fit *sdm_slf2* that modeled SLF suitability from the predicted values of *sdm_toh*. As such, *sdm_slf2* represents suitability that considers a primary plant host (TOH)²⁵ that is also invasive but likely not at equilibrium⁷⁹ and the same abiotic covariates as *sdm_slf1* (*sdm_toh* uses the same covariates).

Reviewer #2 (Remarks to the Author):

The paper attempts to adapt a public-health framework to invasion science (invasive species) “leveraging an increasingly universal risk vocabulary of pandemics” of infective agents such as covid (organism to organism contact) to the spread and impact of invasive species (I suppose pests and weeds) coining the term paninvasion (pan invasion) to describe their spread, establishment, and impact.

The authors presume that a framework for such analyses of invasive species does not exist, and yet for many decades it has been the underlying bases of the field of biological control (i.e., transport (by whatever mechanism be it natural or anthropogenic), establishment (is weather suitable and are hosts available?), and what is the impact potential.

We thank the reviewer for pointing out that biological control also uses invasion process theory. As we stated in the manuscript, we developed the paninvasion severity assessment framework by adapting the US CDC pandemic severity assessment framework to invasion process theory. We now cite several sources from the biological control and invasion biology literature to make this important point explicit.

We have also made it more explicit that we are in no way saying that we have developed the idea that a species must be transported, establish and impact ecological and human systems to be invasive. Indeed, these ideas are old, in part stemming from Darwin, and formalized as invasion process theory and developed in the biological control as the reviewer correctly point out.

What is novel with our work is the merger of public health frameworks with invasion process theory. Several researchers have called for such merger, and we cite those papers. Thus, our work is timely and an important step in

the field of invasion biology and its application to addressing harm caused by pests. To address this comment we augmented Fig. 2, which now contains a new assessment flowchart panel, **d**, and wrote the following:

Lines 39–47: Despite the importance of identifying emerging paninvasives, existing approaches lack a cohesive and universal framework for rapidly assessing and effectively communicating such risk to stakeholders⁷. To address this gap, we developed the paninvasion severity assessment framework by adapting the US CDC pandemic severity assessment framework to invasion process theory, which describes translocations of species in terms of transport, establishment, and impact potentials (Fig. 1, 2)^{2,8–10}. Although invasive species frameworks are increasingly adapted to understand infectious diseases like COVID-19^{11–16}, adapting public-health frameworks to invasion science is novel and leverages an increasingly universal risk vocabulary (Fig. 2)¹⁷.

Lines 211–30: Although the invasion process can be divided into many stages, the paninvasion severity assessment framework focuses on the three main stages most often estimated in invasion risk assessments² and that are analogous to the disease potentials that public-health scientists quantify for pathogens (Fig. 2)⁸. When a pathogen with pandemic risk emerges, public health scientists place it within scaled measures of transmissibility and infectivity (often combined and termed transmissibility), and virulence (clinical severity) to assess its risk^{8,9}. For example, when SARS-CoV-2 emerged during the COVID-19 pandemic, the initial understanding was that different age groups had similar potentials to transmit and become infected (Fig. 2a, y-axis), but different age groups varied in their clinical severity once infected (Fig. 2a, x-axis)^{8,63,64}. To adapt this public-health framework to invasion process theory^{2,65–68}, we equated transmission, infectivity, and virulence potentials of a pathogen across different human populations to the transport, establishment, and impact potentials of an invasive species across different regions (see colored arrows between Fig. 2a and b). For example, in Fig. 2b we placed several hypothetical regions that together indicate strong alignment (i.e., multivariate correlation) among invasion potentials across the regions. In this example, predicted invasion risk (Fig. 2c, x-axis) for these three hypothetical regions is strongly correlated to a measure of their contributions to a global market (Fig. 2c, y-axis), indicating an overall high paninvasion risk.

Paninvasion assessments comprise four steps (Fig. 2d): 1) estimate invasion potentials, 2) calculate alignment of invasion potentials, 3) quantify paninvasion risk, and 4) articulate caveats, which we describe in detail for SLF below and in the SI methods.

Fig. 2. The paninvasion severity assessment framework is adapted from the US CDC pandemic severity assessment framework used to estimate the risk of emerging human pathogens. For pandemics (a), quadrant plots of pathogen transmissibility and infectivity (combined on one axis) vs. pathogen virulence (clinical severity) are used to

compare the risk of a pathogen across different populations or age groups^{8,9,11}. For paninvasions (b), invasion potentials for an emerging regional invasive species are estimated (d Step 1) by equating pathogen transmission with transport potential, infectivity with establishment potential, and virulence with impact potential (follow the arrows) across regions (black circles) to construct quadrant plots that depict their alignment based on multivariate correlations (d Step 2; see Methods). Next, paninvasion risk (c) is estimated from the correlation between regional invasion risk estimated from the multivariate regression of invasion potentials (d Step 3; see Methods) and the size of regional markets that could be disrupted. The steps of the paninvasive severity assessment framework (d), culminating by articulating caveats in the current assessment that direct future research (d Step 4) that provide data to inform the next assessment iteration.

Specifically, they use as their foil for their paninvasion severity assessment framework the spreading of the regional invasive US grape pest, the spotted lanternfly planthopper (*Lycorma delicatula*; SLF), and its assumed potential to “disrupt the global wine” market. “Once established, SLF likely has high global impact potential to wine markets because grape is an equally suitable host”, – i.e., but it is not equally preferred compared to its invasive host *Ailanthus altissima* (tree of heaven) also native to China.

We thank the reviewer for their emphasis on host preference for TOH vs. grape. To be clear, current research shows that SLF has similar preference levels for TOH and grapes, SLF can develop on TOH and grape when provided only a single host, and most importantly, SLF develops with increased fecundity when allowed to feed on both TOH and grape. This research clearly suggests that in landscapes with both TOH and grape—such as in the important viticultural areas of California, France, Italy, and Spain—SLF will likely have high development rates. To address this comment, we added citations and rewrote this passage in the introduction:

Lines 62–6: Once established, SLF likely has high global impact potential to wine markets because grape is an equally suitable host. SLF develops at similar rates when fed grape or TOH, and fecundity increases when fed a mixed diet of these two preferred hosts^{25,30–33}. Asian vineyard production is impacted by SLF infestations^{34,35}; and SLF-invaded US vineyards have reported vine deaths, >90% yield losses, and closure (Fig. 1g–j)^{21,23,36}.

Using the SDM MaxEnt, they found the apparent future distribution of lantern fly is roughly same as that of its preferred host (*A. altissima*) used as component to estimate establishment potential (i.e., worldwide) that in the palearctic is similar to that of grape. Using multivariate statistics (multiple linear regression?), “...We found that SLF invasion potentials are aligned globally because important viticultural regions with suitable environments for SLF establishment also heavily trade with invaded US states”. They state the “Alignment correlations among transport, establishment, and impact potentials were positive for impact potential measured as state grape production ($\rho = 0.41$, $P < 0.001$), etc.”, and yet the meaning of alignment correlations and statistic ρ were not defined. The language is clear, but the meaning is opaque. For example, “Going forward, paninvasion potentials for other species are likely to increasingly align and coordinated governmental efforts will be needed to reduce invasion potentials in the US and internationally”.

We thank the reviewer for suggesting that we clarify our methods for transparency. Much of our methods are in the Supplementary Information. We have altered the passages that discuss pertinent methods and results below for clarity:

Lines 115–29: SLF invasion potentials across states and countries were aligned. Alignment correlations calculated as Spearman’s rank correlations (ρ statistic) among transport, establishment, and impact potentials were positive for impact potential measured as state grape production ($\rho = 0.41$, $P < 0.005$), state wine production ($\rho = 0.52$, $P < 0.001$), country grape production ($\rho = 0.67$, $P < 0.001$), and country wine production ($\rho = 0.63$, $P < 0.001$). This alignment of potentials is clear in the invasion-potential alignment plots (Fig. 5). Major grape producing regions fall in the upper-right quadrant of the plots where regions have both high transport and high establishment potentials.

Lines 317–8: Then, we regressed country grape production on transport and establishment potentials with multiple linear regression.

In summary, the major strength is the MaxEnt analysis (potential for establishment) that has previously been projected by another author using MaxEnt, while the major weakness is the transport mechanism that while likely valid conceptually for SLF is at best a very rough measure, but such measures are likely not valid for other plant/animal species that disperse/invade in a variety of ways.

This is a great point that we addressed in the previous version of the manuscript, and we thank the reviewer for emphasizing it. Like the reviewer stated, our metric of transport potential corresponds well to SLF biology and has a strong relationship with both spread to established areas within the US and regulatory incidents reported to date. SLF has been found hitchhiking on a wide range of commodities and transportation infrastructure. To address this

comment, we have updated our passage from the methods, included reference to the Supplementary Information, and added more details in our caveats section as follows:

Lines 92–3: The current SLF spread in the US (Fig. 3) was explained by this transport potential metric (SI, Supplementary Table 1).

SI Lines 59–74: The prevailing hypothesis on SLF transport potential is that regions that import more tonnage of commodities from the invaded US region also import more total tonnage of goods and trade infrastructure (e.g., cargo containers, pallets, railcars) that inadvertently transport SLF egg masses long-distances. SLF propagules have been found hitchhiking on and in shipments of pharmaceutical containers, baking ingredients, paint shipments, building materials, boxes of pumpkins, pallets and many other commodities^{6,8–12}. To test if total tonnage can explain the current spread of SLF, we fit two logistic regressions with our metric of transport potential based on total tonnage as the covariate. This metric was the \log_{10} of the average annual metric total tonnage imported between 2012 and 2017 from US states invaded by SLF (main text Fig. 3). We regressed the presence/absence of established populations and regulatory incidents (i.e., has a state experienced and reported any observations of SLF, dead, moribund, or alive, independent of the presence of established populations?). For both establishment and regulatory incidents, the relationship between SLF-status and our measure of transport potential was significant, thereby providing support for our estimate of SLF transport potential (Supplemental Table 1). These results suggest that total tonnage of imports is a suitable proxy for transport potential until new metrics are developed that include refined pathway analyses.

Their two caveats to quantifying invasion severity for SLF render mute their global assessments: paraphrased, (1) they estimate invasion severity of SLF via a stepping-stone introduction from the eastern US, but (2) because major wine producing nations also heavily trade with China where SLF is native and Japan and South Korea where SLF is established, SLF transport potential is greater than their estimates, meaning that invasion severity is also likely higher than what we report here, and future work should account for global trade network dynamics.

We thank the reviewer for restating the caveats that we already raised in the manuscript. First, as the reviewer points out, our measure is a conservative estimate of paninvasion risk. Conservative estimates are better than no estimate at all or estimates that are more likely to overstate risk and cause more alarm and waste more resources than is warranted. Second, both the distribution of SLF in China and the import production network from China into other nations are unknown. Thus, an analysis that looks at spread of SLF through an entire global network of trade is not presently feasible without more information provided by the Chinese government and China-wide surveys of SLF. Given that these two issues are likely not to be solved soon, it is vital that we study this fast-spreading pest with information that is currently available. Third, trade between Europe and the US invaded region is strong, thus the stepping-stone introduction from the US to other parts of the globe is an important pathway to study. The introduction of SLF to the US was the first in the western hemisphere, and it opens new avenues for spread into important viticultural regions of Europe.

To address this comment, we now provide a discussion in our caveats section that taking conservative approaches based on the data on hand, like what is done with pathogens with pandemic potential, is better than waiting to get data that will not be available during the window of time available to take the necessary measures to reduce invasion potentials. Our goal in such revisions is to make it clear that implementation of the paninvasion severity assessment framework is aimed at evaluations of emergent pests that may lack the data available for an invasion at equilibrium but that can be updated as new data become available. Here are the relevant updated passages:

Lines 327–32: Paninvasion risk assessments should be performed iteratively as the invasion process continues across regions and responses are mobilized. Early assessments of emergent pests have great utility to support early responses but often come with caveats. The fourth step of the framework is to articulate the caveats of a current assessment to guide future research. These caveats should explicitly consider the assumptions made when estimating invasion potentials and how those potentials and risk could change as the invasion process continues.

Lines 341–53: We calculate paninvasion risk of SLF via stepping-stone transport from the eastern US. However, major wine producing nations also heavily trade with China, Japan, and South Korea, where SLF is also established. Total SLF transport potential is thus greater than our estimates, meaning paninvasion risk is higher than what we report here. Future work should account for global trade network dynamics to other nations with established SLF populations. However, comprehensive surveys are first needed on the distribution of SLF in these other countries and the trade emanating from regions with established populations. Further research should focus on the identification of which industries and commodities are most likely to increase long-distance spread. This requires linking trade dynamic models to phenological models to indicate if high risk trade is occurring at the same time as egg laying, because eggs are the life-stage most likely to be transported long-distances²¹. Additionally, refinement of SLF

propagule pressure dynamics⁶⁹, specifically propagule number and ratio of successful to failed transportation events, can improve estimates of transport potential.

Lines 147–50: We recommend that estimates of SLF transport potential should be updated regularly: as more states become invaded, by matching seasonal trade dynamics to SLF phenology, and by including new information on high transport potential pathways such as rail, landscaping stone, and live tree shipments^{18,21,34}.

As cast, the transference of the notions of pandemics potential contributes little to the substance of the paper. The paper needs to be rewritten with the focus on the methods and data with the summary term paninvasion introduced to characterize its potential for spread widely.

We thank the reviewer for this opinion. We now point to and discuss the several published opinions that argue that the unification of theory and terminology between epidemiology and invasive species risk assessments are important and novel lines of interdisciplinary research. As suggested by the reviewer, we have also rewritten the paper to provide more clarity on methods and revise use of “paninvasion” as a summary term to characterize a pest’s potential to spread widely. Specifically, we:

- include a section of definitions including paninvasion in the Supplementary Information
- modified the introduction to set a definition promptly:

Lines 28–31: For a regional pest to become a globally invasive species that disrupts global markets (i.e., paninvasive), ecological and economic factors that determine the pest’s transport, establishment, and impact potentials must be aligned at the global scale (Fig. 1, SI)².

- cited the below papers that argue for a unification of frameworks such as we propose here to be clear and important goals (see below citations for support):

Hatcher, M. J., Dick, J. T. A. & Dunn, A. M. Disease emergence and invasions. *J. Ecol.* **26**, 1275–1287 (2016).
Ogden, N. H. *et al.* Emerging infectious diseases and biological invasions: A call for a one health collaboration in science and management. *R. Soc. Open Sci.* **6**, 181577 (2019).
Nuñez, M. A., Pauchard, A. & Ricciardi, A. Invasion science and the global spread of SARS-CoV-2. *Trends Ecol. Evol.* **35**, 642–645 (2020).
Simberloff, D., Meyerson, L. & Fefferman, N. Invasive species policy and COVID-19. (2020).
Comizzoli, P., Pagenkopp Lohan, K. M., Muletz-Wolz, C., Hassell, J. & Coyle, B. The Interconnected Health Initiative: A Smithsonian Framework to Extend One Health Research and Education. *Frontiers in Veterinary Science* **8**, (2021).

Figures 5 and 6 are interesting and with clear explanation of methods would make for an interesting paper. Figure 6 includes data for countries that are not wine producers.

Thank you for the comments for clarification of figures. We have edited the described methods behind Figures 5 and 6 in the methods section, including a clarification that some countries export wine despite not producing it:

Lines 307–9: We then visualized these multiple multivariate correlations as quadrant plots following a stakeholder-friendly and approachable format adapted from the pandemic severity assessment framework^{8,9}.

Lines 312–24: To determine if invasion risk for countries corresponds with economic impact to the global wine industry, we investigated the relationship between wine market size and predicted risk of invasion for individual countries. We estimated wine market size for 223 countries (including some that export but do not produce wine) as the value of wine exports corresponding with the years for our trade data (2012–2017, log₁₀ USD) downloaded from the FAOSTAT detailed trade matrix⁴⁶, accessed August 31, 2020. Then, we regressed country grape production on transport and establishment potentials with multiple linear regression. Each predicted value from this regression can be considered an estimate of the risk of SLF to invade and impact a country’s grape production. We rescaled these estimates from 1–10 to create an easily interpreted estimate of risk and then correlated these predicted values to wine export market size. To place overall SLF paninvasion severity on a clear scale for both researchers and stakeholders, we simply rescaled the Pearson correlation from 1–10, so that 1 is a complete negative correlation and 10 is a complete positive correlation between country risk and wine export market size.

Importantly, they suggest more research on biocontrol which is likely the only long-term solution. I hope my comments will be useful in redrafting the paper.

Thank you. They most certainly have been helpful and have caused us to think more deeply about how we present our work and how it builds on previous work from the biological control literature. To emphasize the importance of biocontrol as being likely the only long-term solution, we have included more recent work on biocontrol in the context of both SLF and TOH in the discussion:

Lines 153–70: We suggest eliminating the horticultural sale of TOH; increasing funding for TOH removal; research on cost-effective TOH biocontrol methods^{e.g., 51}; and because SLF are generalists, more research to identify other suitable hosts found in the landscapes surrounding high transport and impact potential locations like railyards and vineyards^{21,30,52}. Finally, reduction to impact potential currently relies on reducing SLF populations with tree-band trapping and broad-spectrum insecticides (e.g., carbamates, organophosphates, pyrethroids, neonicotinoids) that have high nontarget mortality, do not prevent vineyard reinfestations, and often overlap with grape harvest when adults move into vineyards^{18,36,53,54}. Damaged vines can be pruned, but grape yield is reduced⁵⁴, and therefore we suggest more research on long-term control methods, such as trapping technologies that reduce bycatch, host-specific biocontrol agents, and SLF-specific RNAi insecticides to control outbreaks in vineyards and beyond^{21,55–59}.

REVIEWERS' COMMENTS:

Reviewer #1 (Remarks to the Author):

The authors have addressed my comments to a sufficient extent.